# In Vitro Neuroprotective Effect of the Bovine Umbilical Vein Endothelial Cell Conditioned Medium Mediated by Downregulation of IL-1β, Caspase-3, and Caspase-9 Expression

**DOI:** 10.3390/vetsci9020048

**Published:** 2022-01-27

**Authors:** Vinny A. Larasati, Gregorius V. Lembang, Yudy Tjahjono, Sugi Winarsih, Ika Dewi Ana, Hevi Wihadmadyatami, Dwi L. Kusindarta

**Affiliations:** 1Department of Anatomy, Faculty of Veterinary Medicine, Universitas Gadjah Mada, Yogyakarta 55281, Indonesia; vinnyanisya2017@mail.ugm.ac.id (V.A.L.); gregoriusviktor00@mail.ugm.ac.id (G.V.L.); heviwihadmadyatami@ugm.ac.id (H.W.); 2Biomedical Laboratory, Faculty of Pharmacy, Widya Mandala Catholic University, Surabaya 60112, Indonesia; yudy.tjahjono@gmail.com; 3Department of Agriculture, Food, and Fisheries of Sleman Regency, Yogyakarta 55281, Indonesia; wiwin.surat@gmail.com; 4Department of Dental Biomedical Sciences, Faculty of Dentistry, Universitas Gadjah Mada, Yogyakarta 55281, Indonesia; ikadewiana@ugm.ac.id

**Keywords:** BUVEC-CM, neurodegeneration, neuroprotective, apoptosis

## Abstract

Mesenchymal stem cells (MSCs) and conditioned medium (CM) derived from human umbilical blood cord stem cells (HUBSC) are now being extensively utilized. Human umbilical vein endothelial cells (HUVECs) have the same ability as HUBSC as an option for autologous therapy. In addition, cell therapy using HUVECs may produce protective signals for cerebral vessels and promote neuronal survival after hypoxic–ischemic damage. HUVECs have the same anatomical and physiological structure as bovine umbilical vein endothelial cells (BUVECs). In this study, we aim to determine the ability of BUVEC-CM to reduce inflammation and apoptosis on in vitro neurodegeneration models (PC12 and SH-SY5Y cell lines). BUVEC-CM obtained from the third and fourth passages were analyzed using liquid chromatography–mass spectrometry (LC-MS) and high-resolution mass spectrometry (HR-MS), while the other part was used as a treatment for in vitro model neurodegeneration. The PC12 and SH-SY5Y cell lines were cultured and grouped into seven different treatments, including untreated cells. As the treatment group, cells were given TMT 10 µM in the presence of different doses of CM (25%, 50%, 75%, and 100%); as a control comparison of recent therapy, donepezil was used. In addition, cells with the administration of TMT 10 µM were run as a positive control. Cell viability assay (CCK-8) and enzyme-linked immunosorbent assay (ELISA) were performed to identify the viability and expression of interleukin-1β (IL-1β), caspase-3, and caspase-9 for both PC12 and SH-SY5Y cells. The results showed that BUVEC-CM could significantly reduce IL-1β expression and downregulate caspase-3 and caspase-9, as well as when compared to the donepezil group. Taken together, these results indicate that BUVEC-CM can be used as a potential candidate for neuroprotective agents by reducing the activity of IL-1β and the expression of caspase-9 and caspase-3 in PC12 and SH-SY5Y cells induced by TMT. However, further research still needs to be conducted.

## 1. Introduction

Neurodegenerative disease is a pathological condition in the nervous system or neurons characterized by loss of function, structure, or both [1]. Some diseases classified as neurodegenerative diseases include Alzheimer’s disease (AD), Parkinson’s disease, Huntington’s disease, and amyotrophic lateral sclerosis [2]. AD ranks first as the most common neurodegenerative disease affecting people globally, especially the elderly (75–84 years) [3,4]. In 2016, the incidence of AD reached 43,386 cases, and Parkinson’s disease reached 6063 [4,5].

Mesenchymal stem cells (MSCs) are used extensively in tissue repair, growth, wound healing, and cell substitution caused by physiological and pathological conditions. MSCs derived from human umbilical cord blood stem cells (HUCBs) are more widely used to treat neurodegenerative diseases than other types of cell cultures. MSCs are known to proliferate and differentiate into various cell forms, such as adipocytes and chondrocytes. When cultured, HUCB-MSCs will not age with an increase in the number of passages. However, this therapy with stem cells still requires further research on the mechanism of therapy, immune rejection reactions, and the possibility of tumorigenesis [3,4]. Besides therapy with MSCs, using a conditioned medium (CM) derived from MSCs as a cell-free therapy has excellent prospects. Currently, the use of CM has been widely developed in handling cases of various diseases due to its production ease, packaging, and transportation, and because it allows for allogeneic transplantation. However, the lack of clinical trials regarding the use of CM as a disease therapy is a weakness of using CM as a cell-free therapy [6].

Human umbilical vein endothelial cells (HUVECs) and HUCBs have the same potential to be developed as cell therapy. HUVECs have other specific benefits, including cell homogeneity, surface marker characterization, endothelial progenitor identification, and the ability to obtain large cell numbers via rapid expansion. Autologous treatment using HUVECs for newborns is particularly appropriate because the umbilical cord contains a sufficient number of vascular endothelial cells. Through its paracrine effect, cell therapy using HUVECs may produce protective signals for cerebral vessels and promote neuronal survival after hypoxic–ischemic damage in the neonatal brain [7]. Injury or neuronal damage may rapidly promote the expression of proinflammatory cytokines such as interleukin-1β [8]. Shen et al. reported interleukin-1β expressed highly on the degenerative intervertebral disc and nucleus pulposus. In addition, interleukin-1β may induce apoptosis through the mitochondria [9]. Anatomically and physiologically, the umbilical cord of domestic cattle is similar to the umbilical cord of humans, where both the bovine and human umbilicals have two arteries and one vein wrapped in Wharton’s jelly, which also substitutes for the *tunica adventitia* [10,11,12,13]. Therefore, MSCs and CM isolated from the bovine umbilical vein are hypothetically expected to be potential alternatives in regenerative medicine, mainly as neurodegenerative medicine. This study aimed to determine the neuroprotective effect of CM derived from the bovine umbilical vein by reducing the inflammation and apoptosis on in vitro neurodegeneration models (PC12 and SH-SY5Y cell lines).

## 2. Materials and Methods

### 2.1. Materials

Dulbecco’s phosphate-buffered saline (DPBS), Hank’s balanced salt solution (HBSS), fetal bovine serum (FBS), penicillin and streptomycin (P/S), and Accutase were purchased from Capricorn (Ebsdorfergrund, Germany). Dulbecco’s modified Eagle medium (DMEM) and collagenase type II, 125 U, were purchased from Gibco (Langenselbold, Germany). Dimethyl sulfoxides (DMSO) were purchased from Sigma-Aldrich (Munich, Germany). RIPA lysis buffer was purchased from Santa Cruz Biotechnology (Santa Cruz, TX USA). Human interleukin-1β (IL-1β), caspase-3, and caspase-9 ELISA kits were purchased from Fine Test (Wuhan, China). Amphoterycin was purchased from Gibco (Langenselbold, Germany). Donepezil was purchased from PT Etercon Pharma (Demak, Indonesia). Trimethyltin (TMT) was purchased from Sigma (Munich, Germany). CCK-8 was purchased from Abbkine (Wuhan, China).

### 2.2. Cells

Primary cells, bovine umbilical vein endothelial cells (BUVECs), were isolated from a three-year-old Holstein Friesian cow that originated from the farm of the Sleman municipality. The isolation protocol is described below. PC12 and SH-SY5Y were purchased from the European Collection of Authentication Cell Cultures (ECACC). Unless stated otherwise, all the cells used in this manuscript were incubated in a 37 °C incubator with a 5% CO_2_ environment condition.

### 2.3. Isolation, Cultivation, and Cryopreservation of Bovine Umbilical Vein Endothelial Cells (BUVECs)

Approximately 10 cm of bovine umbilical vein was rinsed three times with 10 mL of DPBS. To detach the endothelial cells, the lumen of the bovine umbilical vein was injected and further incubated with 10 mL of collagenase type II (0.025% diluted in HBSS) for 30 min. The detached cells were collected in a 15 mL falcon tube. After subsequent centrifugation (2400 rpm, 5 min, 24 °C), the pellet containing BUVECs was resuspended with a cultivation medium (DMEM supplemented with 10% FBS, 2% P/S, and 0.5% amphotericin). The detached cells were cultivated in a T-25 flask and resuspended with a 7 mL cultivation medium, approximately three days after the cells reached 80% confluency.

For further cultivation, the cells were washed with DPBS, centrifuged (6500 rpm, 5 min, 24 °C), and incubated with 1 mL of Accutase (3 mL for a T-75 flask) for 5 min. The Accutase was inactivated with the addition of an 8-fold (*v*/*v*) cultivation medium. After subsequent washing and centrifugation, the cells were resuspended in 15 mL medium in a T-75 flask for cultivation. This step was repeated for each passage with other clean T-75 flasks. For cryopreservation, 8 × 10^6^ cells were collected in a cryotube, resuspended with 10% DMSO, and kept in a −80 °C refrigerator.

### 2.4. Maintenance of PC12 and SH-SY5Y Cell Lines

The T-25 flask was coated with 2 mL of 0.5% bovine gelatin and further incubated. Gelatin bovine of 0.5% was discharged and replaced with a 5 mL culture medium. The PC12 cell line was thawed (37 °C, 2 min) and later transferred in a 15 mL tube with a 9 mL culture medium (DMEM, 5% horse serum, 1% P/S, 1% amphotericin), and centrifuged (6500 rpm, 5 min, 22 °C). The supernatant was discharged, and the pellet was resuspended with a 3 mL culture medium. The cell was cultured in the previously prepared T-25 flask. The cell was subcultured after it reached 80% confluency. The old culture medium was discharged; then, the flask was washed with 1–2 mL of DPBS and discharged. Accutase of 1 mL was added and then incubated for 5 min. T-25 was tapped to help detach the cell from the T-25 surface. The cell was observed under a microscope to ensure that it was already detached. A 7 mL culture medium was added to inactivate the Accutase and resuspended. Half of the medium was then transferred to another T-25 flask to incubate. Subsequently, for the SHSY-5Y, the cells were maintained on DMEM in the presence of 15% FBS, 1%P/S, 1% amphotericin, 2 mM glutamine, and 1% nonessential amino acids (NEAA). The cells were used maximum on the passage of 18.

### 2.5. Isolation and Analysis of BUVEC-CM with Liquid Chromatography–Mass Spectrometry (LC-MS) and High-Resolution Mass Spectrometry (HR-MS)

The isolated medium from the surface of the BUVEC was defined as “CM”. In summary, after each BUVEC passage, the medium at the surface of the cells was harvested with a 20 mL cannula injector, filtered with a 0.22 µm sterile filter, and kept in a 4° refrigerator for further assay. To analyze the secretome in the BUVEC-CM, LC-MS was performed from the collected BUVEC-CM at the third and fourth passages. The sample in liquid form was taken sufficiently, dissolved in ethanol, and then filtered with a 0.22 µm sterile filter. A 5 mL sterile sample was injected into the LC-MS (Mobile Phase A: water formic acid 0.01%; Mobile Phase B: CH_3_CN, formic acid, 0.01% positive polarity).

Subsequently, HR-MS analysis was also performed to detect the metabolite on the BUVEC-CM. A 5 µL sterile sample was injected into the HR-MS (Mobile Phase A: water and 0.1% formic acid; Mobile Phase B: acetonitrile + 0.1% formic acid). BUVEC-CM Compound identification was run by Thermo Scientific™ Compound Discoverer Software (mzVault, ChemSpider, and mzCloud library version 3.3, Thermo Scientific™, Waltham, MA, USA).

### 2.6. Morphological Assessment of the PC12 and SH-SY5Y Cell Lines

A total of 287 µL aliquots of 1.5 × 10^6^ cells/mL PC12 cells or 193 µL aliquots of 3 × 10^6^ cells/mL SH-SY5Y were transferred to 96-well plates and incubated for 24 h before treatment. During treatment, the cells were preincubated with 10 µM TMT and 50 µL BUVEC-CM (0%, 25%, 50%, 75%, and 100%). The cells were then washed and fixed with 200 µL of 4% paraformaldehyde (PFA; Sigma) for 15 min. The cells were analyzed using inverted microscopy with 40× magnification (Nikon Eclipse TE2000-E, Tokyo, Japan). Three different people observed the morphology of the cell and determined the description of the morphological condition of the cell and the semiquantitative condition of the cellular density.

### 2.7. CCK-8 Cell Viability Assay of PC12 and SH-SY5Y Cell Lines with BUVEC-CM

A total of 287 µL aliquots of 1.5 × 10^6^ cells/mL PC12 cells or 193 µL aliquots of 3 × 10^6^ cells/mL SH-SY5Y were transferred to 96-well plates and incubated for 24 h before treatment. During treatment, the cells were preincubated with 10 µM TMT and 50 µL BUVEC-CM (0%, 25%, 50%, 75%, and 100%) diluted in PC12 cell medium (for CCK-8 assay with PC12) or SH-SY5Y cell medium (for CCK-8 assay with SH-SY5Y), followed by a 6 h incubation for the PC12 cell line and a 24 h incubation for the SH-SY5Y cell line. As a positive control for the cell viability assay in each cell type, cells were incubated with TMT only. As a control for BUVEC-CM activity against caspase activity and proinflammatory cytokines assay, 10 µM of donepezil was given to the cells in addition to TMT. In another well, the cells were left untreated for negative controls. After the 6 h and 24 h incubations, CCK-8 reagent was added, followed by a 4 h incubation, and quantified using 450 nm absorbance with the microplate reader (Tecan 20M, Zurich, Switzerland).

### 2.8. Treatment and Preparation of Cell Lysate and Enzyme-Linked Immunosorbent Assay (ELISA) for the Detection of IL-1β, Caspase-3, and Caspase-9

A total of 1340 µL aliquots of 2.6 × 10^6^ cells/mL PC12 or SH-SY5Y cells were cultivated in a 6-well plate for 24 h. After washing with DPBS, the cells were treated for another 24 h with serial dilution of BUVEC-CM (25%, 50%, 75%, and 100%), 1 µM donepezil, or left untreated. Before lysis of the cells, the washing step was performed again with DPBS, followed by the addition of 900 µL of RIPA lysis buffer. The cells were incubated for 15 min with constant shaking, and the cells were detached with a cell scraper. The cell suspension was centrifuged at 6500 rpm for 10 min at 4 °C, and the supernatant was collected for ELISA. Sandwich ELISA to determine the IL-1β, caspase-3, and caspase-9 concentrations was performed according to the manufacturer’s instructions (Fine Test, Wuhan, China) and quantified using 450 nm absorbance with a microplate reader (Tecan 20M, Zurich, Switzerland).

### 2.9. Statistics

ANOVA statistical analysis, followed by a Bonferroni post hoc test, was performed using GraphPad Prism 9 software (GraphPad Software, San Diego, CA, USA). *p* < 0.05 was considered statistically significant.

### 2.10. Ethical Clearance

The experimental procedures were approved by the Ethical Committee of the Faculty of Veterinary Medicine, Universitas Gadjah Mada, Yogyakarta, Indonesia (Approval Number 00057/EC-FKH/Int./2021).

## 3. Results

### 3.1. LC-MS and HR-MS Analysis of BUVEC-CM

The LC-MS and HR-MS analysis of BUVEC-CM demonstrated the composition of the BUVEC-secretome (Table 1) on the third and fourth passages.

### 3.2. BUVEC-CM Maintains the Morphological Integrity and Cellular Viability of Rat Pheochromocytoma PC12 and Human Neuron-like SH-SY5Y Cell Lines after Treatment with the Neurotoxic Agent TMT

To assess the effect of BUVEC-CM on the morphological integrity of PC12 and SH-SY5Y cell lines, after treatment with 10 µM of the proapoptotic substance TMT, we performed a cytological observation using inverted microscopy. The experiments were repeated during each passage, and each was triplicated until the fifth passage. We observed that the treatment with BUVEC-CM did not alter the integrity and formation of both the PC12 (Figure 1A) and SH-SY5Y (Figure 1B) cell lines. The cells remained adherent and maintained their polygonal shapes. Treatment with 10 µM TMT (in the absence of BUVEC-CM) significantly reduced the cell density, indicating the reduction in cell viability and cytotoxicity induced by TMT. Interestingly, treatment with BUVEC-CM (25–100% dilution) inhibited the reduction in the cell population induced by TMT. There was no significant difference in cellular morphology and cellular density observed between each dilution variant of the BUVEC-CM-treated cell lines.

The morphological assessment of BUVEC-CM in rat pheochromocytoma PC12 and human neuron-like SH-SY5Y cell lines was further revalidated and quantified in a spectrometry-based CCK-8 cell viability assay (Figure 2). Treatment of 1.5 × 10^6^ cells/mL PC12 cells or 3 × 10^6^ cells/mL SH-SY5Y lines with 10 µM TMT (in the absence of BUVEC-CM) significantly reduced the CCK-8 cell viability indexes, represented by the reduction in the cell population, and indicated the cytotoxicity performed by TMT. According to the morphological observation, treatment of both cell lines with TMT and several dilutions of BUVEC-CM did not show a significant reduction in the CCK-8 indexes. The cell populations in all concentration variants (25%, 50%, 75%, and 100%) of BUVEC-CM-treated cell lines, indicated by the CCK-8 indexes, remained high and about the same level when compared with the untreated cell lines (OD = 1.16577 for BUVEC-CM 25% + TMT; OD = 1.1729 for BUVEC-CM 50% + TMT; OD = 1.1919 for BUVEC-CM 75% + TMT, OD = 1.2359 for BUVEC-CM 100% + TMT versus 1.46043 ± SD for untreated PC12 cells; OD = 1.5673 for BUVEC-CM 25% + TMT; OD = 1.6766 for BUVEC-CM 50% + TMT; OD = 1.67993 for BUVEC-CM 75% + TMT, OD = 1.67307 for BUVEC-CM 100% + TMT versus 1.9308 ± SD for untreated SH-SY5Y cells). There was no significant difference in CCK-8 indexes between each dilution variant of the BUVEC-CM-treated cell lines. The treatment with BUVEC-CM in both cell lines maintains cellular viability despite the presence of the proapoptotic TMT substance.

### 3.3. BUVEC-CM Inhibits Proinflammatory Cytokines IL-1β Production of Rat Pheochromocytoma PC12 and Human Neuron-like SH-SY5Y Cell Lines after Induction with the Proapoptotic Substance TMT

To analyze IL-1β production in PC12 and SH-SY5Y after induction of 10 µM TMT and BUVEC-CM treatment, sandwich ELISA was performed on the cytosolic part of the cells (cell lysate). The untreated rat PC12, as well as untreated human SH-SY5Y cells, produced the lowest IL-1β, indicated by the lowest OD (0.1635 in the PC12 cell line and 0.13565 in the SH-SY5Y cell line) generated from both cell lines compared with other treatments (Figure 3). Compared with untreated cells, induction of TMT could significantly generate intracellular IL-1β expression of both cell lines (OD = 0.6716 for PC12 cells, *p* = 0.0008; OD = 0.6061 for SH-SY5Y cells, *p* < 0.0001, compared with cells + TMT). Additional treatment of TMT-treated cells with 1 µM donepezil could diminish intracellular IL-1β expression (OD = 0.21735 for PC12 cells, *p* = 0.0015; OD = 1.48077 for SH-SY5Y cells, *p* < 0.0001, compared with cells + TMT). In PC12 cells, treatment with increased BUVEC-CM after induction of 10 µM TMT shows a significant dose-dependent reduction in IL-1β expression (*p* = 0.0319 for 25% BUVEC-CM; *p* = 0.0230 for 50% BUVEC-CM; *p* = 0.0057 for 75% BUVEC-CM; *p* = 0.0035 for 100% BUVEC-CM, compared with cells + TMT). In SH-SY5 Y cells, treatment with increased BUVEC-CM after induction of 10 µM TMT shows a significant dose-dependent reduction in IL-1β expression (*p* < 0.0001 for 25% and 50% BUVEC-CM; *p* = 0.0002 for 75% BUVEC-CM; *p* < 0.0001 for 100% BUVEC-CM, compared with cells + TMT). Interestingly, treatment with BUVEC-CM, even in the lowest concentration (25%), could diminish IL-1β production in the human neuroblastoma-derived SH-SY5Y cell lines (*p* < 0.0001).

### 3.4. BUVEC-CM Inhibits Proapoptotic Caspase-3 and Caspase-9 Production of Rat Pheochromocytoma PC12 and Human Neuron-like SH-SY5Y Cell Lines Treated with TMT

To investigate the effect of BUVEC-CM on the apoptosis pathway, we performed another sandwich ELISA in the PC12 (Figure 4A) and SH-SY5Y (Figure 4B) cell lysates. The representative proapoptotic parameters observed were caspase-3 and caspase-9. The untreated rat PC12 and untreated human SH-SY5Y cells produced the lowest caspase-3 and caspase-9, indicated by the lowest OD generated from both cell lines when compared with the other treatments. When compared with the untreated cells, the induction of TMT could significantly induce intracellular caspase-3 and caspase-9 expression in both cell lines (*p* = 0.0019 for caspase-3 and *p* < 0.0001 for caspase-9 in the PC12 cell line; *p* = 0.0055 for caspase-3 and *p* = 0.0020 for caspase-9 in the SH-SY5Y cell line). Additional treatment of TMT-treated cells with 1 µM donepezil could diminish the caspase-3 and caspase-9 expression. In PC12 cells, treatment with increased BUVEC-CM after induction of 10 µM TMT showed a static reduction in caspase-3 and caspase-9 expression. Treatment of increased BUVEC-CM after induction of 10 µM TMT could inhibit caspase-3 production in SH-SY5Y cells. Interestingly, the reduction in caspase-9 in BUVEC-CM-treated, TMT-induced SH-SY5Y cells shows a dose-dependent pattern, with the highest caspase-9 expression generated from 100% BUVEC-CM-treated cells.

## 4. Discussion

Neurodegeneration refers to the progressive process of atrophy and the loss of function of neurons, which is present in neurodegenerative diseases, such as Alzheimer’s disease (AD) and Parkinson’s disease (PD). The loss of function of the nervous system causes disorders related to the individual’s learning and memory [1,14]. The therapy for neurodegeneration that has been applied still shows many weaknesses, and currently, there are no medications that can regenerate striatal neurons in PD. Meanwhile, the treatment given for AD is mostly aimed at palliative therapy with a two-strategy approach, namely symptomatic treatment (anticholinesterase inhibitors) and disease-modifying treatment (antioxidants, anti-inflammatory). Various therapies have been used to treat AD, and some of them have even entered clinical trials. For example, immunotherapy using antibodies used to treat amyloid-beta plaques and some antibodies against fibrillar tangles have been used in animal models of AD. Recent clinical trials to treat AD have been attempted by several pharmaceutical companies. However, without satisfactory results, Bapineuzumab, an antibody against Tau that causes fibrillary tangles, was tested by Pfizer, and they reported clinical results from their trial, which found that Bapineuzumab had no benefit for mental function in people with mild to moderate AD [15]. Lilly’s anti-amyloid-beta (solanezumab) failed to improve cognition in AD patients in a phase III clinical trial [16]. There is still no treatment option that can cure amyotrophic lateral sclerosis (ALS) and Huntington’s disease (HD) [15]. Thus, alternative therapies are needed. In this study, we used a CM obtained from BUVECs as a neuroprotectant candidate and as neurodegeneration therapy.

BUVEC-CM at various concentration levels (25%, 50%, 75%, and 100%) is known to maintain the morphological integrity and viability of the PC12 and SH-SY5Y cell lines. At various concentrations, BUVEC-CM also reduced IL-1β, followed by a decrease in the expression of caspase-3 and caspase-9 in the PC12 and SH-SY5Y cell lines. The administration of TMT caused high IL-1β activity and the expression of caspase-3 and caspase-9 in the positive control group. This can be caused by nerve cell damage induced by TMT. When nerve damage occurs, microglia cells become activated and produce several anti-inflammatory and proinflammatory factors that can modulate neurogenesis. Microglial cells can produce proinflammatory cytokines, such as tumor necrosis factor (TNF-α), IL-1β, and IL-6, and reactive oxygen species (ROS), which can inhibit neurogenesis and induce cell apoptosis in progenitor neurons [17,18]. The induction of apoptosis by IL-1β can occur via the mitochondrial pathway or the intrinsic pathway. IL-1β increased the expression of the proapoptotic protein Bax and decreased the antiapoptotic protein Bcl-2. Simultaneously, there was a decrease in cytochrome C in the mitochondria and an increase in the expression of cytochrome C in the cytoplasm. Cytochrome C released in the cytoplasm will activate Apaf1, which together with pro-caspase-9 will induce the formation of apoptosomes, followed by the activation of pro-caspase-3 by caspase-9 [17,19]. It can be seen also in our results that damage to PC12 and SH-SY5Y cells in the presence of TMT is caused by increasing IL-1β activity, thus affecting caspase-9 and caspase-3 activation to cause apoptosis.

Yang et al. [20] suggested that many factors and signaling pathways activated by inflammation are involved in the process of apoptosis. Research on NP cells conducted by Shen et al. [9] proved that IL-1β facilitates the process of apoptosis through the intrinsic or mitochondrial pathways. In this pathway, IL-1β will increase the expression of Bax/Bcl-2, releasing cytochrome C from the mitochondria to the cytoplasm, which will eventually activate caspase-9 and caspase-3, resulting in apoptosis. The same thing happened in a study using cardiomyocytes conducted by Del Re et al. [21], where they showed that IL-1β stimulation also activates the apoptotic pathway intrinsically by releasing cytochrome C in the mitochondria into the cytoplasm. Interestingly, what is the mechanism by which BUVEC-CM decreases IL-1β expression and acts as neuroprotection in neuron cells that experience inflammation due to TMT administration?

To answer this question, we performed a metabolomic analysis using LC-MS and HR-MS. It was found that the CM obtained from BUVECs contained many amino acids and other secondary metabolites. Tjalsma et al. propose the term secretome, a biomolecule that is secreted by cells, tissues, or organisms through various secretory mechanisms [22]. The biomolecules consist of cytokines, adhesion molecules, hormones, growth factors, neurotransmitters, and proteases that describe cell function [22]. In cell culture, the secretome or exosome is secreted into the medium, and secretomes are known to increase the differentiation and proliferation of progenitor neurons in brain regions [23]. Ribeiro et al. [24] conveyed that the conditioned medium of adipose-derived stem cells induces neuritogenesis mediated by nerve growth factor (NGF) stimulation. Neuronal proliferation and differentiation involve growth factors secreted by cells, such as brain-derived neurotrophic factor (BDNF), neuronal growth factor (NGF), vascular endothelial growth factor (VEGF), and fibroblast growth factor 2 (FGF-2) both in vitro and in vivo [25,26] In our study, we conducted an initial study to determine the metabolite components present in the secretome, where we found that using LC-MS and HR-MS, it was known that the secretome contains a lot of amino acids, where these amino acids are needed in the process of secretory protein formation. Secretory proteins themselves are known to have 16–45 amino acids in their signal peptides. Several studies have shown that the amino acid alone may give advantages as a neuroprotectant. Cheng et al. [27] reported L-lysine confers neuroprotection after intracranial hemorrhage injury by reducing inflammatory response and enhancing microglial polarization, mediated by upregulation of miRNA-575 and downregulation PTEN. In addition, glycine confers neuroprotection against D-galactose-induced neurodegeneration and memory impairment in mouse brain [28]. So, we can propose that the amino acid content in BUVEC-CM plays a role in preventing inflammation and apoptosis so that it can function as a neuroprotectant.

Li et al. [29] conveyed that the administration of MSC-CM can prevent apoptosis induced by various neurotoxic substances. It was reported that MSC-CM contains neurotrophic factors such as BDNF and NGF, secreted by MSCs, and these neurotrophic factors may have antiapoptotic effects [30,31]. In addition, endothelial cells are also known to produce neurotrophic factors, such as brain-derived neurotrophic factors and vascular endothelial growth factor. This substance can protect neurons against inflammation and oxidative damage and maintain the integrity of the neovasculature [31,32,33]. Of the various neurotrophic factors, nerve growth factor (NGF) was first isolated and characterized for its ability to regulate and increase neuronal survival. Yang et al. [20] reported that in vitro cell treatment with NGF can reduce the incidence of neuronal cell apoptosis and caspase-3 expression. CM produced by MSCs and HUVECs can downregulate the synthesis of TNF-α, IL-1β, IL-6, and iNOS, as well as caspase-3 [34,35]. The administration of BUVEC-CM at several concentrations informs us that the CM produced by BUVEC can prevent apoptosis mediated by downregulation of IL-1β, caspase-9, and caspase-3 (Figure 5).

## 5. Conclusions

Taken together, it can be concluded that BUVEC-CM was able to reduce the activity of IL-1β and the expression of caspase-9 and caspase-3 in PC12 and SH-SY5Y cells induced by TMT. Therefore, we suggest, BUVEC-CM may serve as a good candidate for the neuroprotectant on the neurodegeneration disease; however, further research is needed to find out more about the safety and dosage of BUVEC-CM.

## Figures and Tables

**Figure 1 vetsci-09-00048-f001:**
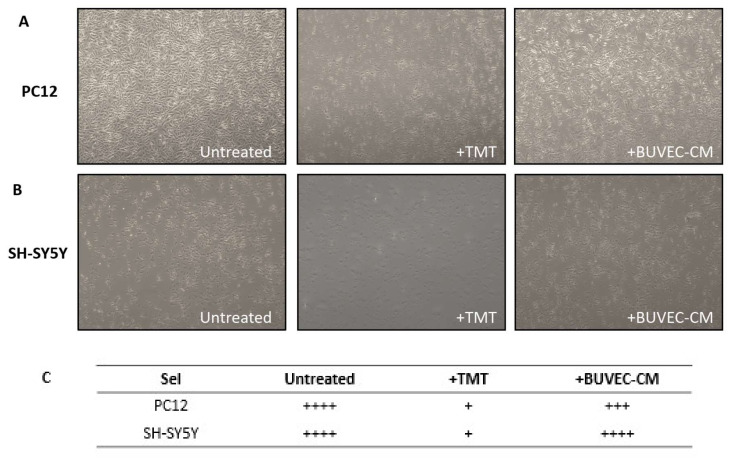
(**A**) Rat pheochromocytoma PC12 cell lines and (**B**) SH-SY5Y cell lines morphology assessment after treatment with BUVEC-CM and neurotoxic agent TMT (**C**) Semiquantitative measurement for the cell density. Compared with untreated cells, treatment with 10 µM TMT (+ TMT, in the absence of BUVEC-CM) significantly reduced the cell density, indicating a reduction in cell viability and cytotoxicity induced by TMT. Interestingly, treatment with BUVEC-CM (+ BUVEC-CM) inhibited the reduction in the cell population induced by TMT. Images were taken with an inverted microscope (40× magnification); + : a few; +++/++++: numerous).

**Figure 2 vetsci-09-00048-f002:**
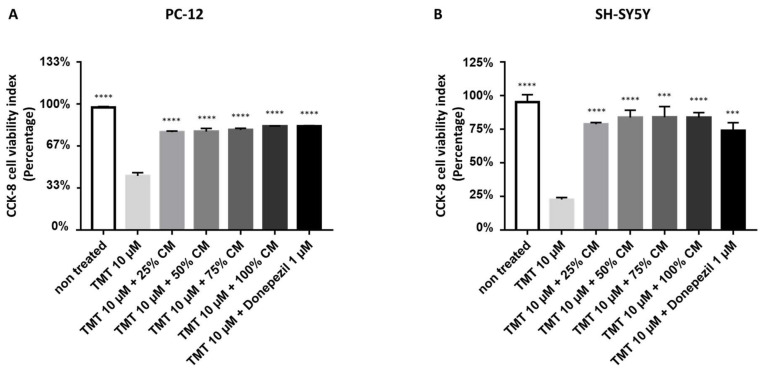
BUVEC-CM maintains the cellular viability of rat pheochromocytoma PC12 and human neuron-like SH-SY5Y cell lines after treatment with the neurotoxic agent TMT. Before adding the CCK-8 cytotoxicity assay, (**A**) rat pheochromocytoma PC12 cell lines and (**B**) human neuron-like SH-SY5Y cell lines were preincubated with 10 µM TMT or equimolar TMT + BUVEC-CM (0%, 25%, 50%, 75%, 100%) diluted in the respective cell line media. Untreated cells were used as the CCK-8 assay control. Data are presented as optical density (OD) proportional to the CCK-8 cell viability indexes. A high OD represents high cell viability. *** *p* < 0.0001, **** *p* < 0.00001 against TMT groups.

**Figure 3 vetsci-09-00048-f003:**
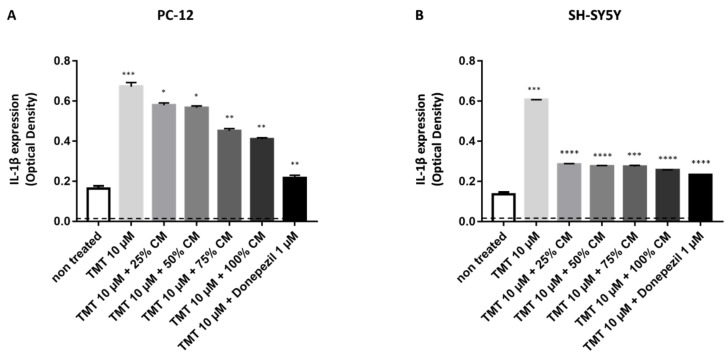
BUVEC-CM inhibits proinflammatory cytokine IL-1β production after treatment with the neurotoxic agent TMT. To induce IL-1β expression, (**A**) rat pheochromocytoma PC12 and (**B**) human neuron-like SH-SY5Y cell lines were preincubated with 10 µM TMT or equimolar TMT + BUVEC-CM (0%, 25%, 50%, 75%, and 100%) diluted in the respective cell line media. Untreated cells were used as a negative control. The analgesic agent donepezil was used as a specific IL-1β inhibitor induced by TMT. A dose-dependent reduction in IL-1β concentration was observed with cytoplasmic lysate from the PC12 and SH-SY5Y cells treated with TMT and increased BUVEC-CM concentration. A constant reduction in IL-1β concentration was observed in the SH-SY5Y cells treated with TMT and increased BUVEC-CM concentration. Data are presented as optical density (OD) proportional to IL-1β concentration. A high OD represents high IL-1β production. * *p* < 0.01, ** *p* < 0.001, *** *p* < 0.0001, **** *p* < 0.00001 against untreated groups.

**Figure 4 vetsci-09-00048-f004:**
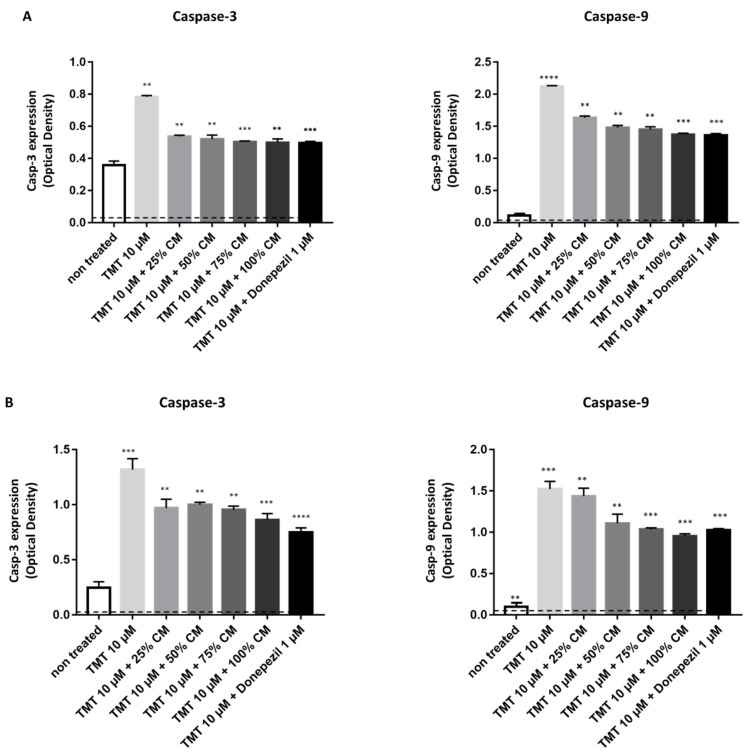
BUVEC-CM inhibits proapoptotic caspase-3 and caspase-9 production of rat pheochromocytoma PC12 and human neuron-like SH-SY5Y cell lines treated with TMT. To induce apoptosis, (**A**) rat pheochromocytoma PC12 and (**B**) human neuron-like SH-SY5Y cell lines were preincubated with 10 µM TMT or equimolar TMT + increased BUVEC-CM concentration (25–100%) diluted in the respective cell line media. Untreated cells were used as a negative control. The analgesic agent Donepezil was used as a specific apoptosis inhibitor induced by TMT. A reduction in the apoptosis parameters, caspase-3 and caspase-9, was observed with ELISA of cytoplasmic lysate from PC12 and SH-sy5y cells treated with TMT and BUVEC-CM concentrations. Particularly, a dose-dependent reduction in caspase-9 concentration was observed in SH-SY5Y cells treated with TMT and increased BUVEC-CM concentration. Data are presented as optical density (OD) proportional to both caspase concentrations. A high OD represents high caspase production. ** *p* < 0.001, *** *p* < 0.0001, **** *p* < 0.00001 against untreated groups.

**Figure 5 vetsci-09-00048-f005:**
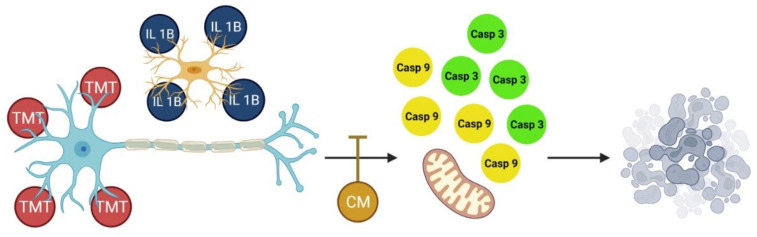
Through the in vitro neurodegenerative models presented in this manuscript, BUVEC-CM reduced the activity of IL-1β and the expression of caspase-9 and caspase-3 in PC12 and SH-SY5Y cells induced by the neurotoxic agent TMT. This observation indicates the potential of BUVEC-CM as a neuroprotectant agent, particularly in neurodegenerative diseases.

**Table 1 vetsci-09-00048-t001:** Primary metabolites present in bovine umbilical mesenchymal stem cells CM generated by LC-MS and HR-MS.

Number	Metabolite
1	Isoleucine
2	Leucine
3	Valine
4	Ethanol
5	Threonine
6	Lactate
7	Alanine
8	Lysine
9	Arginine
10	Acetate
11	Glutamate
12	Glutamine
13	Methionine
14	Hydroxyproline
15	Hydroxyurasile
16	Choline
17	Pruvate
18	a-Glucose
19	b-Glucose
20	Tyrosine
21	Phenylalanine
22	Histidine
23	Tryptophan
24	Palmitic acid
25	Ethyl palmitoleate
26	Stearic acid
27	Cholecalciferol
28	Oleamide
29	Ethyl myristate
30	Nicotinamide
31	Format

## Data Availability

The data presented in this study are available in the manuscript.

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
