# Peer review of "In Vitro Neuroprotective Effect of the Bovine Umbilical Vein Endothelial Cell Conditioned Medium Mediated by Downregulation of IL-1β, Caspase-3, and Caspase-9 Expression"

_vetsci, 2022, doi:10.3390/vetsci9020048_

Round 1
Reviewer 1 Report
Dear authors, the manuscript needs extensive editing of english. Here below only two little corrections at the beginning of the abstract. Moreover, some paragraphs lack fluidity and there are errors like double spaces, no capital letter after the point, and so on. Please ameliorate the english structure and grammar of the manuscript with an english editing service.
Abstract: Utilization of mesenchymal stem cells (MSCs) and conditioned medium (CM) derived 18 from human umbilical blood cord stem cells (HUBSC) has now been widely applied.
after hypoxic-ischemic --> after hypoxic ischemic injury
Introduction: MSCs derived from human umbilical cord blood stem cells (HUCB) are more widely used to 50 treat neurodegenerative diseases than other types of cell culture (this affirmation lacks a reference).
Figure 5: Please enlarge the image, it's difficult to read the content of the circles. The caption does not fully explain the image.
Discussion: Neurodegeneration is a disease that attacks the nervous system and generally causes 289 disorders related to the learning and memory of the individual.
Neurodegeneration is a condition... or Neurodegenerative diseases attack... or Neurodegeneration refers to the progressive atrophy and loss of function of neurons (https://www.nature.com/subjects/neurodegeneration)
It's deeply incorrect to refer to neurodegeneration as a "disease".
290: The therapy for neurodegeneration that has been applied still shows many weaknesses, so alternative therapies are needed.
The therapy for what? For amyloid deposition? for tau? for lewy bodies formation?
Author Response
Reviewer 1:
Dear authors, the manuscript needs extensive editing of english. Here below only two little corrections at the beginning of the abstract. Moreover, some paragraphs lack fluidity and there are errors like double spaces, no capital letter after the point, and so on. Please ameliorate the english structure and grammar of the manuscript with an english editing service.
Abstract: Utilization of mesenchymal stem cells (MSCs) and conditioned medium (CM) derived 18 from human umbilical blood cord stem cells (HUBSC) has now been widely applied.
after hypoxic-ischemic --> after hypoxic ischemic injury
Introduction: MSCs derived from human umbilical cord blood stem cells (HUCB) are more widely used to 50 treat neurodegenerative diseases than other types of cell culture (this affirmation lacks a reference).
Answer: thank you very much for the reviewer valuable suggestion we already do the English manuscript editing according to the author suggestion to the sribendi (https://www.scribendi.com), and revise the abstract as the reviewer suggestion in the revision manuscript.
Figure 5: Please enlarge the image, it's difficult to read the content of the circles. The caption does not fully explain the image.
Answer: thank you very much for the reviewer valuable suggestion we already enlarged the figure 5, as the reviewer suggestion in the revision manuscript
Discussion: Neurodegeneration is a disease that attacks the nervous system and generally causes 289 disorders related to the learning and memory of the individual.
Neurodegeneration is a condition... or Neurodegenerative diseases attack... or Neurodegeneration refers to the progressive atrophy and loss of function of neurons (https://www.nature.com/subjects/neurodegeneration)
It's deeply incorrect to refer to neurodegeneration as a "disease".
Answer: thank you very much for the reviewer valuable suggestion we already revise the sentence according to the reviewer suggestion in the revision manuscript, on the lane 316-319 and we mark it in red
“ Neurodegeneration refers to the progressive process of atrophy and the loss of function of neurons, which is present in neurodegenerative diseases, such as AD and Parkinson’s disease. The loss of function of the nervous system causes disorders related to the individual’s learning and memory.
290: The therapy for neurodegeneration that has been applied still shows many weaknesses, so alternative therapies are needed.
The therapy for what? For amyloid deposition? for tau? for lewy bodies formation?
Answer: thank you very much for the reviewer valuable suggestion we already revise the sentence according to the reviewer suggestion in the revision manuscript, on the lane 319-333 and we mark it in red
The therapy for neurodegeneration that has been applied still shows many weaknesses, and currently, there are no medications that can regenerate striatal neurons in Parkinson’s disease. Meanwhile, the treatment given for AD is mostly aimed at palliative therapy with a two-strategy approach, namely symptomatic treatment (anticholinesterase inhibitors) and disease-modifying treatment (anti-oxidants, anti-inflammatory). Various therapies have been tried to treat AD, and some of them have even entered clinical trials. For example, immunotherapy using antibodies used to treat amyloid beta plaques and some antibodies against fibrillar tangles has been tried in animal models of AD. Recent clinical trials to treat AD have been attempted by several pharmaceutical companies. However, without satisfactory results, Bapineuzumab, an antibody against Tau that causes fibrillary tangles, was tested by Pfizer, and they reported clinical results from their trial, which found that Bapineuzumab had no benefit for mental function in people with mild to moderate AD. Eli Lilly’s anti-amyloid beta (Solanezumab) failed to improve cognition in AD patients in a Phase III clinical trial. There is still no treatment option that can cure ALS and HD (Kiaei, 2013);
Kiaei M. New hopes and challenges for treatment of neurodegenerative disorders: great opportunities for young neuroscientists. Basic Clin Neurosci. 2013 Winter;4(1):3-4. PMID: 25337322; PMCID: PMC4202552.

Reviewer 2 Report
Ref: vetsci-1494925
Title: In-vitro neuroprotective effect of the Bovine Umbilical Vein Endothelial Cell Conditioned Medium mediated by down-regulation of IL-1β, Caspase-3 and Caspase-9-expression
Recommendation: Major Revision/Reject
Overview:
The paper is focused on neuroprotective properties of the bovine umbilical vein endothelial cell conditioned medium (BUVEC-CM). The research was conducted on PC12 and SH-SY5Y cell lines. The Authors used CCK-8 cell viability assay, ELISAs and mass spectrometry (LC-MS) to verify the research hypothesis.
My biggest objection to this paper is based on the use of PC12 and SH-SY5Y cells without providing any information about differentiation of cells into neurons. For that reason, it can be suspected that the research was carried out on undifferentiated cells, and thus, we cannot speak of neurotoxicity in this case.
Major comments:
- The Authors do not provide any information about differentiation of cells into neurons. Differentiation of cells lines such as PC12 and SH-SY5Y into neurons takes several days and is the most important step. Since there is no information about the differentiation, it can be suspected that the research was carried out on undifferentiated cells, and thus, we cannot speak of neurotoxicity in this case.
- The Part 2.4. – maintenance of PC12 and SH-SY5Y cell lines should be describe in details.
Minor comments:
- The correct name of cell line is SH-SY5Y.
- Please provide the aim of the study at the end of the Introduction part.
- Fig. 2A – why there is **** sign above the non-treated cells and no significance sign above TMT 10 uM?
- Fig. 2B – why there is **** sign above the non-treated cells and no significance sign above TMT 10 uM?
- Fig. 3A – no SE/SEM above the non-treated cells.
- Fig. 3B – no SE/SEM above the non-treated cells.
- Fig. 4A and Fig. 4B - caspase-9 – no SE/SEM above the non-treated cells.
Author Response
Reviewer 2
The paper is focused on neuroprotective properties of the bovine umbilical vein endothelial cell conditioned medium (BUVEC-CM). The research was conducted on PC12 and SH-SY5Y cell lines. The Authors used CCK-8 cell viability assay, ELISAs and mass spectrometry (LC-MS) to verify the research hypothesis.
My biggest objection to this paper is based on the use of PC12 and SH-SY5Y cells without providing any information about differentiation of cells into neurons. For that reason, it can be suspected that the research was carried out on undifferentiated cells, and thus, we cannot speak of neurotoxicity in this case.
Major comments:
- The Authors do not provide any information about differentiation of cells into neurons. Differentiation of cells lines such as PC12 and SH-SY5Y into neurons takes several days and is the most important step. Since there is no information about the differentiation, it can be suspected that the research was carried out on undifferentiated cells, and thus, we cannot speak of neurotoxicity in this case.
Answer: Thank you very much for the reviewer valuable suggestion and comment, The following is an explanation from us regarding the cells we use:
- PC12 cell line: originally the cells comes from ECAAC, we have used these cells for research on the group of one member of the researcher (Dr. Ika Dewi Ana). Previous studies which is done by PC12 already maintained the cells with nerve growth factor (NGF) and differentiates into cells that resemble biochemically and phenotypically sympathetic nerves, the following has been done by one of our research teams where from this publication we have known and analyzed that in PC12 cells express achetilcholinesterase (AChE):
Ardhani, R., Ana, I. D., & Tabata, Y. (2020). Gelatin hydrogel membrane containing carbonate hydroxyapatite for nerve regeneration scaffold. Journal of Biomedical Materials Research Part A. doi:10.1002/jbm.a.37000
We did not perform analyzes specifically for these cells because we have no interest in aiming at a specific phenotype but only focusing on neurons.
- SH-SY5Y cell line: originally the cells comes from ECAAC (ECACC catalog no. 94030304) and based on the cell line profile (https://www.phe-culturecollections.org.uk/media/114601/sh-sy5y-cell -line-profile.pdf) its mention “Both undifferentiated and differentiated SH-SY5Y have been reported to express dopaminergic neuronal markers and muscarinic and nicotinic adrenergic receptors. Differentiated SH-SY5Y may be driven towards an adrenergic phenotype” Because we didn't focus on one particular type of neuron so we didn't differentiate further for SHSY-5Y but we took care not to use these cells beyond passage 18
- The Part 2.4. – maintenance of PC12 and SH-SY5Y cell lines should be describe in details.
Answer: thank you very much for the reviewer valuable suggestion, as the reviewer suggestion we already revise the part of “maintenance of PC12 and SH-SY5Y cell lines” in the revision manuscript and mark it in red
Minor comments:
- The correct name of cell line is SH-SY5Y.
Answer: Thank you very much for the reviewer valuable correction, we already revise the name of the cell line as the reviewer suggestion in revision manuscript.
- Please provide the aim of the study at the end of the Introduction part.
Answer: Thank you very much for the reviewer valuable correction we already add the aim of the study in the introduction part on the revision manuscript.
- Fig. 2A – why there is **** sign above the non-treated cells and no significance sign above TMT 10 uM?
Answer: Thank you very much for the reviewer valuable correction, we already add SE/SEM above the non-treated cells in the revision manuscript.
- Fig. 2B – why there is **** sign above the non-treated cells and no significance sign above TMT 10 uM?
Answer: Thank you very much for the reviewer valuable correction, we already add SE/SEM above the non-treated cells in the revision manuscript.
- Fig. 3A – no SE/SEM above the non-treated cells.
Answer: Thank you very much for the reviewer valuable correction, we already add SE/SEM above the non-treated cells in the revision manuscript.
- Fig. 3B – no SE/SEM above the non-treated cells.
Answer: Thank you very much for the reviewer valuable correction, we already revise the figure by adding SE/SEM above the non-treated cells in the revision manuscript.
- Fig. 4A and Fig. 4B - caspase-9 – no SE/SEM above the non-treated cells.
Answer: Thank you very much for the reviewer valuable correction, we already revise the figure by adding SE/SEM above the non-treated cells in the revision manuscript.

Reviewer 3 Report
Dear Editor/Authors
I send you the comments of the work titled “In-vitro neuroprotective effect of the Bovine Umbilical Vein 2 Endothelial Cell Conditioned Medium mediated by down-regulation of IL-1β, Caspase-3 and Caspase-9-expression”.
I have certain comments that, in my opinion, would improve the manuscript.
Abstract
1.- LC-MS is the acronym.
2.- CCK-8 indicate that is a cell viability assay.
3.- Line 30: . In addition,
4.- Line 35: the conclusion of the abstract does not match with the conclusion of the paper.
Introduction:
1.- References should be indicated by number in the text. Check all the references, they are not correctly cited along the text; ack of reference in the sentence 48-49 of the second page, for example.
2.- Indicate the reference for the first sentence and I should take off “or neurons” because glial cells are also included in nervous system.
3.- Indicate a reference for the second sentence also.
4.- The incidence of AD and PD is worldwide? It should be indicated.
5.- Lines 66-67 of the second page, remove the comma: “Autologous treatment using HUVECs for newborns is particularly appropriate because”
6.- Please, rewrite the hypothesis.
Material and Methods
1.- Please indicate the procedure to obtain the bovine umbilical vein.
2.- Cells. Indicate the origin of PC12 (rat pheochromocytoma PC12) here not in the results section.
3.- Correct: “Cells were” instead of “cell was” or “cells was”.
4.- Anova = ANOVA (line 163). Which post hoc test was used?
Results
1.- What does it means particularly after 3rd and 4th passage?
2.- It should be better to indicate the percentage of each metabolite of the secretome.
3.- Correct: Lines 179-180: To assess the effect of BUVEC-CM on morphological integrity of PC12 and SH-SY5Y cell lines after TMT treatment…
4.- They must use an image analyser software to analyse the morphology of cells.
5.- They must indicate cellular density results.
6.- How they can analyse cell morphology in a cell viability assay?
7.- Figure 2. It should be easier to understand if the non-treated group index was 1 or 100 and the rest % of control.
8.- I should recommend to merge sections 3.2 and 3.2. Indicate the densities, then indicate cell viability results.
9.- Indicate how many experiments have done.
- Figure 3 and 4 same as Figure 2.
11.- Why do you analyse IL-1Beta? Indicate in the introduction, not in the discussion.
Discussion
1.- Neurodegeneration is a disease? Not only learning and memory.
2.- Do they really think that they can use as therapy?
3.- They are not co-culturing cells with microglia, or do they?
4.- The do not answer the question of the mechanism. First of all, they do not know if CM contains BDNF or NGF.
5.- Overall, the discussion is very poor and they do not discuss their results.
Author Response
Reviewer 3
Dear Editor/Authors
I send you the comments of the work titled “In-vitro neuroprotective effect of the Bovine Umbilical Vein 2 Endothelial Cell Conditioned Medium mediated by down-regulation of IL-1β, Caspase-3 and Caspase-9-expression”.
I have certain comments that, in my opinion, would improve the manuscript.
Abstract
1.- LC-MS is the acronym.
Answer: thank you very much for the reviewer valuable suggestion, we already add the full abbreviation of LC-MS and put the acronym on the bracket in the abstract section on the revision manuscript and we mark it in red
2.- CCK-8 indicate that is a cell viability assay.
Answer: thank you very much for the reviewer valuable correction, we already put the sentence of cell viability assay replacing the CCK-8 on the abstract section on the revision manuscript and we mark it in red
3.- Line 30: . In addition,
Answer: thank you very much for the reviewer valuable correction we already add the word “In addition,….” on the revision manuscript in the abstract section and mark it in red
4.- Line 35: the conclusion of the abstract does not match with the conclusion of the paper.
Answer: thank you very much for the reviewer valuable suggestion, and as the reviewer suggestion we already revise the conclusion on the abstract section and make it match with the conclusion on the paper in the revision manuscript and we mark it in red.
Introduction:
1.- References should be indicated by number in the text. Check all the references, they are not correctly cited along the text; ack of reference in the sentence 48-49 of the second page, for example.
Answer: thank you very much for the reviewer valuable correction, we already put the references indicated by number in text and make sure that all the reference already cite on the text.
2.- Indicate the reference for the first sentence and I should take off “or neurons” because glial cells are also included in nervous system.
Answer: thank you very much for the reviewer valuable correction and as the reviewer correction we already ad the reference for the first sentence as:
Dugger, B. N., & Dickson, D. W. (2017). Pathology of Neurodegenerative Diseases. Cold Spring Harbor Perspectives in Biology, 9(7), a028035.doi:10.1101/cshperspect.a028035
3.- Indicate a reference for the second sentence also.
Answer: thank you very much for the reviewer valuable correction and as the reviewer correction we already ad the reference for the second sentence as:
Ruffini, N., Klingenberg, S., Schweiger, S., & Gerber, S. (2020). Common Factors in Neurodegeneration: A Meta-Study Revealing Shared Patterns on a Multi-Omics Scale. Cells, 9(12), 2642.doi:10.3390/cells9122642
4.- The incidence of AD and PD is worldwide? It should be indicated.
Answer: thank you very much for the reviewer valuable correction and as the reviewer correction we already ad the reference for the third sentence as:
Deuschl, G., Beghi, E., Fazekas, F., Varga, T., Christoforidi, K. A., Sipido, E., … Feigin, V. L. (2020). The burden of neurological diseases in Europe: an analysis for the Global Burden of Disease Study 2017. The Lancet Public Health, 5(10), e551–e567. doi:10.1016/s2468-2667(20)30190-0
Feigin, V. L., Nichols, E., Alam, T., Bannick, M. S., Beghi, E., Blake, N., … Ellenbogen, R. G. (2019). Global, regional, and national burden of neurological disorders, 1990–2016: a systematic analysis for the Global Burden of Disease Study 2016. The Lancet Neurology.doi:10.1016/s1474-4422(18)30499-x
5.- Lines 66-67 of the second page, remove the comma: “Autologous treatment using HUVECs for newborns is particularly appropriate because”
Answer: thank you very much for the reviewer valuable correction, and as the reviewer valuable correction we already remove the comma on the sentence in the revision manuscript and we mark it in red.
6.- Please, rewrite the hypothesis.
Answer: Thank you very much for the reviewer valuable suggestion, and as the reviewer suggestion we already revise the hypothesis on the revision manuscript in the introduction section and we mark it in red.
“Anatomically and physiologically, the umbilical cord of domestic cattle is similar to the umbilical cord of humans, where both the bovine and human umbilicals have two arteries and one vein wrapped in Wharton’s jelly, which also substitutes for the tunica adventitia [7,8]. Therefore, MSCs and CM isolated from bovine umbilical are hypothetically expected to be potential alternatives in regenerative medicine, mainly as neurodegenerative medicine. This study was aimed to determine the neuroprotective effect of CM derived from bovine umbilical by reducing the inflammation and apoptosis on in vitro neurodegeneration models (PC12 and SH-SY5Y cell lines).”
Material and Methods
1.- Please indicate the procedure to obtain the bovine umbilical vein.
Answer: Thank you very much for the reviewer valuable suggestion, we already add the procedure to obtain the bovine umbilical vein endothelial cell in the material and methods as part 2.3. Isolation, Cultivation, and Cryopreservation of Bovine Umbilical Vein Endhotelial Cell (BUVEC), in the revision manuscript and we mark it in red.
2.3. Isolation, Cultivation, and Cryopreservation of Bovine Umbilical Vein Endhotelial Cell (BUVEC)
Approximately 10 cm bovine umbilical was rinsed three times with 10 ml DPBS. To detach the endothelial cells, the lumen of the bovine umbilical was injected and further incubated with 10 ml of Collagenase Type II (0.025% diluted in HBSS) for 30 min. The detached cells were collected in a 15 ml falcon tube. After subsequent centrifugation (2400 rpm, 5 min, 24°C), the pellet containing BUVEC was resuspended with a cultivation medium (DMEM supplemented with 10% FBS, 2% P/S, and 0.5% Amphoterycin). The detached cells were cultivated in T-25 flask and resuspended with a 7 ml cultivation medium, approximately after three days after the cells reached 80% confluency. For further cultivation, the cells were washed with DPBS, centrifuged (6500 rpm, 5 min, 24 °C), and incubated with 1 ml Accutase (3 ml for a T-75 flask) for 5 min. The Accutase was inactivated with the addition of an 8-fold (v/v) cultivation medium. After subsequent washing and centrifugation, the cells were resuspended in a 15 ml medium in a T-75 flask for cultivation. This step was repeated for each passage with other clean T-75 flasks. For cryopreservation, 8 × 106 cells were collected in a cryotube, resuspended with 10% DMSO, and kept in a −80°C refrigerator.
2.- Cells. Indicate the origin of PC12 (rat pheochromocytoma PC12) here not in the results section.
Answer: thank you very much for the reviewer valuable correction, we already add the origin and explanation about the PC12 (rat pheochromocytoma PC12) on the material method section in the revision manuscript.
3.- Correct: “Cells were” instead of “cell was” or “cells was”.
Answer: thank you very much for the reviewer valuable correction we already revise into “cells were” in the revision manuscript.
4.- Anova = ANOVA (line 163). Which post hoc test was used?
Answer: thank you very much for the reviewer valuable question, for the analysis we use Bonferroni as post hoc test, we already put the type of statistic analysis on the revision manuscript and mark it in red.
“ANOVA statistical analysis, followed by a Bonferroni post hoc test, was performed using GraphPad Prism 9 software (GraphPad Software). P < 0.05 was considered statistically significant”
Results
1.- What does it means particularly after 3rd and 4th passage?
Answer: thank you very much for the very useful question from the reviewer, for this experiment we used the conditioned medium that we obtained in passage three and passage four, because in passage three and passage four, bovine umbilical cell endothelial cells grow optimally to achieve their confluency values and also in this passage, cells grow relatively quickly, around three days to reach confluence compared to passages one and two, in the revision of the manuscript we have changed the sentence as follows:
“The LC-MS and HR-MS analysis of BUVEC-CM demonstrated the composition of the BUVEC-secretome (Table 1), on the third and fourth passages.”
And we also marked this sentence in red in the manuscript revision.
2.- It should be better to indicate the percentage of each metabolite of the secretome.
Answer: Thank you very much for the reviewer valuable suggestion, we conducted a new experiment using HR-MS but we were not able to indicate the percentage of each metabolite due to the limitations of our tool but we added data from the HR-MS test results to our data in the revision manuscript as Table 1.
3.- Correct: Lines 179-180: To assess the effect of BUVEC-CM on morphological integrity of PC12 and SH-SY5Y cell lines after TMT treatment…
Answer: Thank you very much for the reviewer valuable correction, we already revise the sentence as follow according to reviewer correction.
“To assess the effect of BUVEC-CM on the morphological integrity of rat pheochromocytoma PC12 and human neuron-like SH-SY5Y cell lines, after treatment with 10 μM of the proapoptotic substance TMT, we performed a histological observation using inverted microscopy.”
We mark it in red on the revision manuscript.
4.- They must use an image analyser software to analyse the morphology of cells.
Answer:Thank you very much for the reviewer valuable suggestion, we have added the method for determining cell morphology in the material and section method, in the manuscript revision, at lane 147-154, and we use only analysis by microscope viewer.
5.- They must indicate cellular density results.
Answer: Thank you very much for the reviewer valuable suggestion, we have added the method for determining cell morphology in the material and section method, in the manuscript revision, at lane 147-154, and we use only analysis by microscope viewer, to described the density we use semiquantitatively analysis by three person judgment.
6.- How they can analyse cell morphology in a cell viability assay?
Answer:Thank you very much for the reviewer valuable suggestion, we have added the method for determining cell morphology in the material and section method, in the manuscript revision, at lane 147-154, and we use only analysis by microscope viewer.
7.- Figure 2. It should be easier to understand if the non-treated group index was 1 or 100 and the rest % of control.
Answer: thank you very much for the reviewer valuable suggestion we already revise the graphic as the reviewer suggestion (Figure 2) in the revision manuscript.
8.- I should recommend to merge sections 3.2 and 3.2. Indicate the densities, then indicate cell viability results.
Answer: thank you very much for the reviewer valuable suggestion, dan seperti saran dari reviewer we already merge the section 3.2. and 3.3. as section 3.2 as follow:
3.2. BUVEC-CM Maintains the Morphological Integrity and cellular viability of Rat Pheochromocytoma PC12 and Human Neuron-like SH-SY5Y Cell Lines After Treatment With the Neurotoxic Agent TMT
We mark the sentence in red on the revision manuscript as lane 182-184
9.- Indicate how many experiments have done.
Answer: Thank you very much for the reviewer valuable suggestion we do each experiment for triplicate and we maintained until fifth passage to observed the morphology and cell density
To assess the effect of BUVEC-CM on the morphological integrity of PC12 and SH-SY5Y cell lines, after treatment with 10 μM of the proapoptotic substance TMT, we performed a cytological observation using inverted microscopy. The experiments were repeated during each passage each are triplicated until the fifth passage.
- Figure 3 and 4 same as Figure 2.
Answer: thanks for the reviewer valuable suggestion, and as the reviewer valuable suggestion we can provide the following clarification for figures 3 and 4 we are a bit difficult to make it in percent because the value of optical density indicates the value of how much the approximate expression of each gene is so we still choose to display the data with optical density and add a cut off value.
11.- Why do you analyse IL-1Beta? Indicate in the introduction, not in the discussion.
Answer: thank you very much for the reviewer valuable suggestion, and as the reviewer suggestion we already add the explanation why we analyse the IL-1beta on the introduction section in the revision manuscript and we mark it in red on the revision manuscript.
Discussion
1.- Neurodegeneration is a disease? Not only learning and memory.
Answer: thank you very much for the reviewer valuable suggestion, as the reviewer suggestion we already add more explanation on the revision manuscript in the lane 300-317 on the revision manuscript and we mark it in red.
2.- Do they really think that they can use as therapy?
Answer: Thank you very much for the reviewer valuable suggestion, based on our research we discovered the potential of this BUVEC-CM to inhibit inflammation and also decrease apoptosis in vitro model of neurodegeneration induced by TMT, and indeed further study is needed.
3.- They are not co-culturing cells with microglia, or do they?
Answer: Thank you very much for the reviewer valuable question, in this experiment we did not use co-culturing with microglia, we performed an experiment the cells in the presence of BUVEC-CM on the several concentration.
4.- The do not answer the question of the mechanism. First of all, they do not know if CM contains BDNF or NGF.
Answer: Thank you very much for the reviewer valuable question, and as the reviewer reviewer valuable question we can explain as follow:
Tjalsma et al. Conveying the term secretome, namely as a biomolecule that is secreted by cells, tissues, or organisms through various secretory mechanisms. Biomolecules consist of cytokines, adhesion molecules, hormones, growth factors, neurotransmitters, and proteases that describe cell function (Karagianis et al., 2010). In cell culture, the secretome or exosome is secreted into the medium, which is called the conditioned medium. Secretomes are known to increase the differentiation and proliferation of progenitor neurons in brain regions (Salgado et al., 2015). Ribeiro et al. conveyed that the conditioned medium of adipose-derived stem cells induces neuritogenesis mediated by nerve growth factor (NGF) stimulation. Neuronal proliferation and differentiation involves growth factors secreted by cells, such as brain-derived neurotrophic factor (BDNF), Neuronal Growth Factor (NGF), vascular endothelial growth factor (VEGF), fibroblast growth factor 2 (FGF-2) both in vitro and in vivo (Texeira). In our study, we conducted an initial study to determine the metabolite components present in the secretome, where we found that using LC-MS and HR-MS it was known that the secretome contains a lot of amino acids, where these amino acids are needed in the process of secretory protein formation. Secretory proteins themselves are known to have 16-45 amino acids in their signal peptides, so for the time being we can postulate that these amino acids are precursors for the formation of secretory proteins that are expressed in the secretome in the form of adhesion molecules, hormones, or growth factors.
However, it is indeed and indeed that further studies are needed considering that we have not succeeded in analyzing the protein in the secretions due to the limited time for resolving the objections and the unfinished situation of the covid-19 pandemic, which greatly disrupts laboratory regulations.
Tjalsma H, Bolhuis A, Jongbloed JD, Bron S, van Dijl JM. Signal peptide-dependent protein transport in Bacillus subtilis: a genome-based survey of the secretome. Microbiol Mol Biol Rev. 2000;64(3):515–547.
Karagiannis GS, Pavlou MP, Diamandis EP. Cancer secretomics reveal pathophysiological pathways in cancer molecular oncology. Mol Oncol. 2010;4(6):496–510.
Salgado AJ, Sousa JC, Costa BM, Pires AO, Mateus-Pinheiro A, Teixeira FG, et al. Mesenchymal stem cells secretome as a modulator of the neurogenic niche: basic insights and therapeutic opportunities. Front Cell Neurosci. 2015;9(249):249.
Ribeiro CA, Fraga JS, Grãos M, Neves NM, Reis RL, Gimble JM, et al. The secretome of stem cells isolated from the adipose tissue and Wharton jelly acts differently on central nervous system derived cell populations. Stem Cell Res Ther. 2012;3(3):18.
Teixeira FG, Carvalho MM, Sousa N, Salgado AJ. Mesenchymal stem cells secretome: a new paradigm for central nervous system regeneration? Cell Mol Life Sci. 2013;70(20):3871–3882.
5.- Overall, the discussion is very poor and they do not discuss their results.
Answer: thank you very much for the valuable comments and suggestions from reviewers , we have revised the entirety of our discussion, rewrite again and add new references on the revision manuscript.

Round 2
Reviewer 2 Report
The Authors answered all my questions / comments. In my opinion, the paper is ready for publication.
Author Response
The Authors answered all my questions / comments. In my opinion, the paper is ready for publication.
Answer: thank you very much for valuable suggestions, corrections and also questions from the reviewer, so that we can improve the quality of our manuscripts

Reviewer 3 Report
Dear Authors,
Thank you very much for answering most of my requests. However, there are some point you should improved in the current version of your manuscript.
In the introduction you write Interleuikin 1b and in a long the rest of the manuscript IL-1β.
The results are still poor since they are jugments more than data.
In the discussion, the first paragraph only contains one reference and maybe they should indicate Parkinson's disease as acroninm and in the introduction specify the acronims of Huntington’s disease, and amyotrophic lateral sclerosis.
It is too speculative to think that a conditioned medium can delay the course of a neurodegenerative disease when it has been seen that cell therapy based on mesenchymal cells has not been able to do so. For example, in ALS in this meta-analysis (10.1038/s41536-021-00131-5). Thus, in my opinion, the discussion should be redirected to discuss what you have observed and not directed to cure neurodegenerative diseases.
Best regards
Author Response
In the introduction you write Interleuikin 1b and in a long the rest of the manuscript IL-1β.
Answer: Thank you very much for the reviewer valuable suggestion, we already uniform the term and changed into “Interleukin-1β (IL-1β)”on the introduction section in the revision manuscript
The results are still poor since they are judgments more than data.
Answer: Thank you very much for the reviewer valuable suggestion to the results section, and as the reviewer suggestion we already revised our results and do additional experiment to convince our data analysis mainly on the content of the conditioned medium by using HR-MS, and we also done semiquantitative analysis for cellular density data or viability
In the discussion, the first paragraph only contains one reference and maybe they should indicate Parkinson's disease as acroninm and in the introduction specify the acronims of Huntington’s disease, and amyotrophic lateral sclerosis.
Answer: Thank you very much for the reviewer valuable suggestion and correction, we already revise the first paragraph and put the additional reference on the revision version of the manuscript and we already mark the sentence or changes in red
It is too speculative to think that a conditioned medium can delay the course of a neurodegenerative disease when it has been seen that cell therapy based on mesenchymal cells has not been able to do so. For example, in ALS in this meta-analysis (10.1038/s41536-021-00131-5). Thus, in my opinion, the discussion should be redirected to discuss what you have observed and not directed to cure neurodegenerative diseases.
Answer: thank you very much for the reviewer valuable suggestion, we have already reviewed and revised our discussion in the revision manuscript and hopefully can meet the expectations of the reviewer to be able to discuss based on what we have on the results of our research.
